# Ratio Trace Formulation of Wasserstein Discriminant Analysis

**Hexuan Liu, Yunfeng Cai, You-Lin Chen, Ping Li**
Cognitive Computing Lab
Baidu Research
No.10 Xibeiwang East Road, Beijing 100193, China
10900 NE 8th St. Bellevue, Washington 98004, USA
{lhxuan93, yunfengcai09, cyoulin.tw, pingli98}@gmail.com

## Abstract

We reformulate the Wasserstein Discriminant Analysis (WDA) as a ratio trace problem and present an eigensolver-based algorithm to compute the discriminative subspace of WDA. This new formulation, along with the proposed algorithm, can be served as an efficient and more stable alternative to the original trace ratio formulation and its gradient-based algorithm. We provide a rigorous convergence analysis for the proposed algorithm under the self-consistent field framework, which is crucial but missing in the literature. As an application, we combine WDA with low-dimensional clustering techniques, such as K-means, to perform subspace clustering. Numerical experiments on real datasets show promising results of the ratio trace formulation of WDA in both classification and clustering tasks.

## 1 Introduction

Wasserstein Discriminant Analysis (WDA) [13] is a supervised linear dimensionality reduction technique that generalizes the classical Fisher Discriminant Analysis (FDA) [16] using the optimal transport distances [41]. Many existing works [44, 29, 11, 4] have addressed the issue that FDA only considers global information. In particular, [49] proposed a new formula relaying on worst-case distance; [37] developed a localized version of FDA; [22] provided an adaptive method for learning local structure from data. The recently proposed WDA [13] has the advantage of adaptively capturing both local and global information, and shows competitive performance in classification tasks compared to other supervised dimensionality reduction techniques.

WDA as developed in [13] used the trace ratio formulation to maximize the ratio of the inter-class's regularized Wasserstein distances to the intra-class's regularized Wasserstein distances. Formally, they aimed to solve $\max_{\mathbf{P}} \text{Trace}(\mathbf{P}^T \mathbf{C}_b(\mathbf{T})\mathbf{P})/\text{Trace}(\mathbf{P}^T \mathbf{C}_w(\mathbf{T})\mathbf{P})$ where $\mathbf{C}_b$ and $\mathbf{C}_w$ are the inter-class and intra-class covariance matrices, respectively, and are functions of the optimal transport matrix $\mathbf{T}$. The optimal transport matrix $\mathbf{T}$ quantifies how important the distance between two samples should be in order to obtain a good projection matrix $\mathbf{P}$. The authors in [13] derived the gradient of the objective function with respect to $\mathbf{P}$ and also utilized automatic differentiation to compute the gradients. The difficulties of their approach are 1) the optimization objective is non-convex and non-smooth; and 2) $\mathbf{C}_b$ and $\mathbf{C}_w$ are functions of $\mathbf{T}$ and $\mathbf{T}$ is an implicit function on $\mathbf{P}$. Thus WDA is a bi-level optimization problem [8] and requires solving an optimal transport problem in every step of gradient descent. Due to these complications, theoretical guarantees on the convergence are lacking. Vanilla gradient descent gets stuck easily in the non-smooth region, especially for real datasets, due to the natural structure of the data such as low rank or sparsity. In practice, the approach introduced in [13] can be sensitive to initialization and may take many iterations or even fail to reach convergence. All these issues raise concerns when WDA is applied to real data.

In this paper, we circumvent the aforementioned challenges by reformulating WDA as a ratio trace problem, which has a closed-form solution and can be solved by the generalized eigenvalue decomposition if $\mathbf{T}$ is given. For algorithms of dimensionality reduction, it is common to use ratio trace formulation to approximate trace ratio problems [46]. For example, in Fisher Discriminant Analysis (FDA), these two formulations are both defined and are both served as criterion to maximize inter-class distance while minimizing intra-class distance [15]. Although there are many comparisons between these two formulations when the inter-class and intra-class covariance matrices are fixed [43, 28, 17, 27], they do not concern with the case of the covariance matrices being functions of the discriminative subspace as in WDA. We give numerical comparisons between these two formulations in terms of classification accuracy on simulated as well as real data, on which the proposed formulation either is comparable or outperforms the original formulation.

Specifically, we solve the ratio trace problem: $\operatorname{argmax}_{\mathbf{P}} \operatorname{Trace}(\mathbf{P}^T \mathbf{C}_b(\mathbf{T})\mathbf{P}(\mathbf{P}^T \mathbf{C}_w(\mathbf{T})\mathbf{P})^{-1})$ instead of the original WDA formulation: $\operatorname{argmax}_{\mathbf{P}} \operatorname{Trace}(\mathbf{P}^T \mathbf{C}_b(\mathbf{T})\mathbf{P})/\operatorname{Trace}(\mathbf{P}^T \mathbf{C}_w(\mathbf{T})\mathbf{P})$. We propose an algorithm: **WDA-eig**, to solve the ratio trace problem using the self-consistent field iteration (SCF), and establish a convergence analysis for the general SCF framework with specific application to the WDA context. The SCF iteration was originally used for solving Kohn-Sham equation arising in electronic structure calculations [7]. Most works on SCF concern with the standard eigenvalue problem [47, 24, 6], while convergence analysis for the generalized eigenvalue problem has not appeared in current literature. Our numerical examples demonstrate that the algorithm based on SCF iteration usually converges within a few iterations in practice and is less sensitive to initialization compared to the original approach. We also give a convergent analysis under the SCF framework, which not only provides convergence guarantee to the ratio trace WDA problem but is also applicable to other eigenvector-dependent generalized eigenvalue problem.

As an application, we extend **WDA-eig** to unsupervised clustering. Since WDA requires class labels to calculate the inter- and intra-class Wasserstein distances, a natural solution is to combine WDA with low-dimensional clustering techniques, which requires iteratively applying WDA given updated label information. The new algorithm has a fast convergence compared to the original approach and aid in iteratively applying WDA to find the most discriminative subspace. Several methods [10, 48, 42] that are closely related to our work leverage label information by combining FDA with Kmeans. Our numerical experiments show that the WDA-Kmeans has promising performance compared to existing subspace clustering techniques on real-world datasets.

**Our contribution** in this paper is three-fold. First, we present a ratio trace formulation of the WDA problem. Second, we propose to solve the problem using the SCF iteration, and provide a convergent analysis for the SCF framework as well as specific application to the WDA context. Last but not least, we iteratively apply WDA and low-dimensional clustering technique to perform clustering. We emphasize that we do not attempt solving the original trace ratio formulation of WDA with the proposed algorithm. A better solution to the original formulation is not the focus of this paper.

**Notations.** We use $\|\cdot\|$ to denote the 2-norm of a matrix or vector. $\mathbf{I}_n$ is used to denote the identity matrix of order $n$. For any matrix $\mathbf{X}$, let $x_i$ denote its $i$th column vector and $x_{i,j}$ denote the $(i,j)$th entry. For any $\mathbf{A}, \mathbf{B} \in \mathbb{R}^{m \times n}$, $\langle \mathbf{A}, \mathbf{B} \rangle$ is the inner product of $\mathbf{A}$ and $\mathbf{B}$, i.e., $\langle \mathbf{A}, \mathbf{B} \rangle = \operatorname{trace}(\mathbf{A}^T \mathbf{B})$. Let $\mathbb{S}^n = \{A \in \mathbb{R}^{n \times n} | A = A^T\}$ be the set of symmetric matrices. For a symmetric matrix pair $(\mathbf{A}, \mathbf{B}), \mathbf{A}, \mathbf{B} \in \mathbb{S}^n$ with $\mathbf{B}$ being positive definite, we denote the generalized eigenvalues of $(\mathbf{A}, \mathbf{B})$ by $\lambda_{\min}(\mathbf{A}, \mathbf{B}) = \lambda_n(\mathbf{A}, \mathbf{B}) \leq \cdots \leq \lambda_1(\mathbf{A}, \mathbf{B}) = \lambda_{\max}(\mathbf{A}, \mathbf{B})$. Let $\mathbb{O}^{d \times p}$ represent the set of orthogonal $d \times p$ matrices, i.e., $\mathbb{O}^{d \times p} = \{\mathbf{A} \in \mathbb{R}^{d \times p} \mid \mathbf{A}^T \mathbf{A} = \mathbf{I}_d\}$.

## 2 Methodology

In this section, we first review the existing supervised WDA problem and its gradient-based solver, and reformulate the problem as a nonlinear generalized eigenvalue problem. We then present an algorithm that solves the problem using the self-consistent field iteration.

### 2.1 Background

Wasserstein distance (also known as the optimal transport distance, earth mover distance) is a distance between probability measures that preserves the underlying geometry of the space based on principles from the optimal transport theory [41]. The regularized Wasserstein distance is the solu-

tion of the following entropy-smoothed optimal transport problem:

$$\mathbf{T}_\lambda \triangleq \operatorname*{argmin}_{\mathbf{T}\in\mathbb{U}_{nm}} \lambda\langle\mathbf{T},\mathbf{M}_{\mathbf{X},\mathbf{Z}}\rangle - \Omega(\mathbf{T}), \tag{1}$$

where $\lambda \geq 0$ is the Wasserstein regularization parameter, $\mathbf{M}_{\mathbf{X},\mathbf{Z}}$ denotes the pairwise squared Euclidean distance matrix between samples in $\mathbf{X} \in \mathbb{R}^{n\times d}$ and $\mathbf{Z} \in \mathbb{R}^{n\times d}$: $\mathbf{M}_{\mathbf{X},\mathbf{Z}} \triangleq [\|x_i - z_j\|_2^2]$, and $\Omega(\mathbf{T})$ is the entropy of $\mathbf{T}$: $\Omega(\mathbf{T}) \triangleq -\sum_{ij} t_{ij}\log(t_{ij})$. $\mathbb{U}_{mn}$ is the polytope of $m \times n$ nonnegative matrices with row and column sums being equal to $\mathbf{1}_m/m$ and $\mathbf{1}_n/n$ respectively: $\mathbb{U}_{mn} \triangleq \{\mathbf{T} \in \mathbb{R}_+^{m\times n} \mid \mathbf{T}\mathbf{1}_n = \mathbf{1}_m/m, \mathbf{T}^T\mathbf{1}_m = \mathbf{1}_n/n\}$.

As the entropy-smoothed optimal transport problem is strictly convex, the solution to (1) exists and is unique. Numerically, $\mathbf{T}_\lambda$ can be obtained very efficiently using algorithms such as the Sinkhorn's fixed-point iterations [18, 9], the Greenkhorn algorithm [2, 1], or APDAMD [23]. The regularization parameter $\lambda$ can be used to balance the global (at the distribution scale) and the local (at the samples' scale) interactions between different classes.

The original Wasserstein Discriminant Analysis solves the following bi-level optimization problem:

$$\max_{\mathbf{P}\in\mathbb{O}^{d\times p}} J(\mathbf{P}, \mathbf{T}(\mathbf{P})) = \frac{\sum_{c,c'>c}\langle\mathbf{P}\mathbf{P}^T,\mathbf{C}^{c,c'}\rangle}{\sum_c\langle\mathbf{P}\mathbf{P}^T,\mathbf{C}^{c,c}\rangle} = \frac{\langle\mathbf{P}\mathbf{P}^T,\mathbf{C}_b\rangle}{\langle\mathbf{P}\mathbf{P}^T,\mathbf{C}_w\rangle}, \tag{2}$$

$$\text{where } \mathbf{C}^{c,c'} = \sum_{i,j} t_{i,j}^{c,c'}(x_i^c - x_j^{c'})(x_i^c - x_j^{c'})^T, \quad \forall c,c',$$

$$\text{s.t. } \mathbf{T}^{c,c'} = \arg\min_{\mathbf{T}\in U_{n_c n_{c'}}} \lambda\langle\mathbf{T},\mathbf{M}_{\mathbf{X}^c\mathbf{P},\mathbf{X}^{c'}\mathbf{P}}\rangle - \Omega(\mathbf{T}),$$

where $\mathbf{X}^c \in \mathbb{R}^{n_c\times d}$ is the data matrix of the samples from class $c$, and $\mathbf{X}^c\mathbf{P}$ is the matrix of projected samples from class $c$. $\mathbf{C}_b = \sum_{c,c'>c}\mathbf{C}^{c,c'}$ and $\mathbf{C}_w = \sum_c\mathbf{C}^{c,c}$ are the between and within cross-covariance matrices, and they both depend on $\mathbf{T}(\mathbf{P})$.

In [13], the gradient $G^k = \nabla_{\mathbf{P}}J(\mathbf{P},\mathbf{T}(\mathbf{P}))$ at iteration $k$ was computed using automatic differentiation [25], and the optimization problem is solved using pymanopt solvers such as the projected gradient descent and trust region methods on the Stiefel manifold [3, 39]. In practice, due to the complication of the problem formulation and the structures of data, the gradient-based approach often has a slow convergence and is sensitive to parameters and initialization. We will illustrate these difficulties in Section 4 with numerical experiments.

## 2.2 The Nonlinear Eigensolver-based Approach

For (2), once $\mathbf{T}^{c,c'}$ is computed, the problem becomes a trace ratio problem:

$$\max_{\mathbf{P}\in\mathbb{O}^{d\times p}} J(\mathbf{P}) = \frac{\operatorname{Trace}(\mathbf{P}^T\mathbf{C}_b\mathbf{P})}{\operatorname{Trace}(\mathbf{P}^T\mathbf{C}_w\mathbf{P})}, \tag{3}$$

where $\mathbf{C}_b$ and $\mathbf{C}_w$ depend on $\mathbf{P}$. We approximate the problem by solving a ratio trace problem:

$$\max_{\mathbf{P}\in\mathbb{R}^{d\times p}} J_{rt}(\mathbf{P}) = \operatorname{Trace}((\mathbf{P}^T\mathbf{C}_b\mathbf{P})(\mathbf{P}^T\mathbf{C}_w\mathbf{P})^{-1}), \tag{4}$$

Problem (4) can be efficiently solved by the generalized eigenvalue decomposition:

$$\mathbf{C}_b(\mathbf{P})\mathbf{P} = \mathbf{C}_w(\mathbf{P})\mathbf{P}\Lambda, \tag{5}$$

where the optimal $\mathbf{P}$ is the matrix of eigenvectors corresponding to the $p$ largest generalized eigenvalues. The generalized eigenvector-dependent nonlinear eigenvalue problem (which we refer to as NLEP from now on) can be solved via the self-consistent field (SCF) iteration [26, 32]: given $\mathbf{P}_{t-1}$, we first construct $\mathbf{C}_b(\mathbf{P}_{t-1})$ and $\mathbf{C}_w(\mathbf{P}_{t-1})$, then solve the generalized eigenvalue problem $\mathbf{C}_b(\mathbf{P}_{t-1})v = \mu\mathbf{C}_w(\mathbf{P}_{t-1})v$. Let $v_j$ be the eigenvector corresponding to the $j$th largest generalized eigenvalue, then $\mathbf{P}_t$ is updated as an orthonormal basis for $[v_1, \ldots, v_p]$. Compared to the gradient-based approach, the new formulation with SCF iteration could drastically reduce the number of iterations. We therefore propose Algorithm 1 for solving supervised WDA.

---

**Algorithm 1** WDA-eig algorithm

---

**Input:** De-meaned data $X$, class labels $\hat{y}$, initial subspace $\mathbf{P}_0 \in \mathbb{O}^{d \times p}$, tolerance $\epsilon$,
      maximum number of iterations $N$

**for** $k = 1$ **to** $N$ **do**
    **for** each pair of classes $c$, $c'$ **do**
        Compute $\mathbf{T}^{c,c'}(\mathbf{P}_{k-1})$ using the Sinkhorn iteration
    **end for**
    Construct $\mathbf{C}_b(\mathbf{P}_{k-1})$ and $\mathbf{C}_w(\mathbf{P}_{k-1})$
    Compute the generalized eigenvalue problem: $\mathbf{C}_b(\mathbf{P}_{k-1})\mathbf{P} = \mathbf{C}_w(\mathbf{P}_{k-1})\mathbf{P}\Lambda$, and obtain
        $\mathbf{P}_k \in \mathbb{O}^{d \times p}$ as an orthonormal basis for the eigenvector matrix corresponding to the
        $p$ largest generalized eigenvalues
    **if** the change in $\mathbf{P}_k$ is sufficiently small **then**
        Break
    **end if**
**end for**

---

From a computational complexity point of view, suppose that for each class there are $n$ samples and $d$ features. For each SCF iteration, complexity is dominated by constructing $\mathbf{C}_b$ and $\mathbf{C}_w$, which are $\mathcal{O}(n^2 d^2)$. Solving the generalized eigenvalue problem has complexity $\mathcal{O}(d^3)$, but it is possible to only run a few iteration to reach certain tolerance. Each Sinkhorn iteration is of $\mathcal{O}(n^2)$ and we run a fixed number of iterations. The memory complexity is $\mathcal{O}(d^2)$ by storing the matrices $\mathbf{C}_b$ and $\mathbf{C}_w$.

Note that it is also interesting to investigate if the Riemannian optimization can be applied to Problem (4), similarly to [45]. This is, however, nontrivial as the constraint manifold varies with iterations in this case. We leave it to future work. Another line of interesting future work is to develop kernelized version of **WDA-eig** and randomized algorithms to speed up the computations [20, 21]. Moreover, in the future one can also re-visit **WDA-eig** by considering sparsity constraints [5].

## 3 Analysis

In this section we first give a convergence analysis for the SCF framework for solving generalized NLEP, followed by an analysis for the proposed **WDA-eig** in Algorithm 1.

### 3.1 Convergence of SCF

Consider the generalized NLEP $A(\mathbf{P})\mathbf{V} = B(\mathbf{P})\mathbf{V}\Lambda$, where $\mathbf{V} = [v_1, \ldots, v_p]$ and $\mathbf{P}$ is an orthonormal basis of $\mathbf{V}$ that spans the same subspace as $\mathbf{V}$. $A(\mathbf{P}), B(\mathbf{P})$ are symmetric matrix-valued function and $B(\mathbf{P})$ is positive definite. $\Lambda = \text{diag}(\lambda_1, \ldots, \lambda_p)$, where $\lambda_1 \geq \cdots \geq \lambda_p$ are the $p$ largest eigenvalues of $(A(\mathbf{P}), B(\mathbf{P}))$ corresponding to eigenvectors $v_1, \ldots, v_p$. We emphasize that $A(\mathbf{P}), B(\mathbf{P})$ are invariant to orthogonal transformation of $\mathbf{P}$, i.e., $A(\mathbf{P}) \equiv A(\mathbf{P}Q)$, $B(\mathbf{P}) \equiv B(\mathbf{P}Q)$ for any orthogonal matrix $Q \in \mathbb{R}^{p \times p}$.

**Definitions.** Let $\mathcal{X}$ and $\mathcal{Y}$ be two $p$-dimensional subspaces of $\mathbb{R}^n$. Let the columns of $X$ form an orthonormal basis for $\mathcal{X}$ and the columns of $Y$ form an orthonormal basis for $\mathcal{Y}$. We use $\|\sin\Theta(\mathcal{X}, \mathcal{Y})\|$ as in [35] to measure the distance between $\mathcal{X}$ and $\mathcal{Y}$, where

$$\Theta(\mathcal{X}, \mathcal{Y}) = \text{diag}(\theta_1(\mathcal{X}, \mathcal{Y}), \ldots, \theta_p(\mathcal{X}, \mathcal{Y})). \tag{6}$$

Here, $\theta_j(\mathcal{X}, \mathcal{Y})$'s denote the *canonical angles* between $\mathcal{X}$ and $\mathcal{Y}$ [p. 43][35], which is defined as

$$0 \leq \theta_j(\mathcal{X}, \mathcal{Y}) \triangleq \arccos \sigma_j \leq \frac{\pi}{2} \quad \text{for } 1 \leq j \leq k, \tag{7}$$

where $\sigma_j$'s are the singular values of $X^T Y$. Similar to the Crawford number for symmetric definite matrix pair $(A, B)$ [Chapter 8.7] [40], we define the Crawford number for the generalized NLEP as

$$c \triangleq \min_{\mathbf{P} \in \mathbb{O}^{d \times p}} \min_{x \in \mathbb{C}^d, \|x\|=1} (x^T (A(\mathbf{P}) + iB(\mathbf{P}))x),$$

where $i$ is the imaginary unit. Define $C \triangleq \max_{\mathbf{P} \in \mathbb{O}^{d \times p}} \sqrt{\|A(\mathbf{P})^2 + B(\mathbf{P})^2\|}$. At the $k$th SCF iteration, one computes an approximation to the eigenvector matrix $\mathbf{V}_k$ associated with the $p$ largest eigenvalues of $(A(\mathbf{P}_{k-1}), B(\mathbf{P}_{k-1}))$, where $\mathbf{P}_{k-1}$ is an orthonormal basis for $\mathbf{V}_{k-1}$, and then $\mathbf{V}_k$ is used as the next approximation to the solution. Let $A_k = A(\mathbf{P}_k)$, $B_k = B(\mathbf{P}_k)$, $\mu_{i,k} = \arctan \lambda_i(A(\mathbf{P}_k), B(\mathbf{P}_k))$. We also define $s_k \triangleq \|\sin \Theta(\mathbf{P}_k, \mathbf{P}_{k-1})\|$ as the distance between subspaces $\mathbf{P}_k$ and $\mathbf{P}_{k-1}$.

We study the convergence of SCF iteration under the following assumptions:

**A1:** For any $\mathbf{P}_1, \mathbf{P}_2 \in \mathbb{O}^{d \times p}$, assume that there exist positive constants $\xi_a, \xi_b$ such that

$$\|A(\mathbf{P}_1) - A(\mathbf{P}_2)\| \leq \xi_a \|\sin \Theta(\mathbf{P}_1, \mathbf{P}_2)\|, \qquad \|B(\mathbf{P}_1) - B(\mathbf{P}_2)\| \leq \xi_b \|\sin \Theta(\mathbf{P}_1, \mathbf{P}_2)\|;$$

**A2:** For $k = 1, 2, \cdots$, there exists an $\eta > 0$ such that

$$\mu_{p,k} - \mu_{p+1,k} \geq \eta.$$

We state the convergence theorems below and give proofs in the supplementary material. By global convergence we mean that the algorithm converges to some stationary points [34] and does not guarantee convergence to a global optimum for all initial points. The algorithm converges when the change in subspace is sufficiently small, i.e., $s_k$ is within some user-specified tolerance.

**Theorem 1.** *(Global Convergence)* Let $s_1 = \|\sin \Theta(\boldsymbol{P}_0, \boldsymbol{P}_1)\|$. *Assume A1 and A2, and* $s_1 \sqrt{\xi_a^2 + \xi_b^2} < c$. *If*

$$\eta > \arcsin(\rho C \sqrt{\xi_a^2 + \xi_b^2}/c^2) + \arctan(s_1 \sqrt{\xi_a^2 + \xi_b^2}/c)$$

*for some constant $\rho > 1$, then SCF converges linearly at the rate of $\frac{1}{\rho}$.*

With relaxed assumption on the arctangent gap, we can show local convergence if the initial subspace is close enough to the true subspace $\mathbf{P}^*$:

**A3:** Let $\mu_i^*$ denote $\arctan \lambda_i(A(\mathbf{P}^*), B(\mathbf{P}^*))$. There exists an $\eta > 0$ such that

$$\mu_p^* - \mu_{p+1}^* \geq \eta.$$

**Theorem 2.** *(Local Convergence)* Let $\hat{s}_0 = \|\sin \Theta(\boldsymbol{P}_0, \boldsymbol{P}^*)\|$. *Assume A1 and A3, and* $\hat{s}_0 \sqrt{\xi_a^2 + \xi_b^2} < c$. *If*

$$\eta > \arcsin(\rho C \sqrt{\xi_a^2 + \xi_b^2}/c^2) + \arctan(\hat{s}_0 \sqrt{\xi_a^2 + \xi_b^2}/c)$$

*for some constant $\rho > 1$, then SCF is locally convergent at $\boldsymbol{P}^*$ at the rate of $\frac{1}{\rho}$.*

Theorems 1 and 2 characterize how the eigenspace varies when the matrix pair undergoes a small perturbation. The sensitivity of the matrix pair as functions of $\mathbf{P}$ is quantified by the Lipschitz constants in A1. A2 and A3 are assumptions to guarantee that a discriminative subspace exists. In the following section we give more concrete examples in the WDA context for these assumptions.

## 3.2 Analysis for Supervised WDA

In the context of WDA, $A(\mathbf{P})$ is the inter-class covariance matrix $\mathbf{C}_b(\mathbf{P})$ and $B(\mathbf{P})$ is the intra-class covariance matrix $\mathbf{C}_w(\mathbf{P})$. For each iteration in **WDA-eig**, a fixed number of Sinkhorn iterations is computed to obtain an approximation to the optimal transport distance $\mathbf{T}$. $\mathbf{T}(\mathbf{P})$ can be expressed as an implicit function using the optimality conditions of the equation defining the optimal $\mathbf{T}$, and $\frac{\partial \mathbf{T}}{\partial \mathbf{P}}$ exists and is bounded. Therefore it is safe to assume that $\mathbf{T}$ is Lipschitz continuous in $\mathbf{P}$.

**Corollary 1.** *Suppose that the optimal transport matrix $\boldsymbol{T}^{c,c'}$ satisfies a Lipschitz-like condition:*

$$\|\boldsymbol{T}^{c,c'}(\boldsymbol{P}_1) - \boldsymbol{T}^{c,c'}(\boldsymbol{P}_2)\| \leq \xi^{c,c'} \|\sin \Theta(\boldsymbol{P}_1, \boldsymbol{P}_2)\|,$$

*For a given $p$, let*

$$\eta = \min_k \{\eta_k | \eta_k = \mu_{p,k} - \mu_{p+1,k}\}.$$

*Denote $\xi_a = \sum_{c,c'>c} \xi^{c,c'} \| \sum_{i,j} (x_i^c - x_j^{c'})(x_i^c - x_j^{c'})^T \|$, $\xi_b = \sum_c \xi^{c,c} \| \sum_{i,j} (x_i^c - x_j^c)(x_i^c - x_j^c)^T \|$. If*

$$\eta > \arcsin(\rho C \sqrt{\xi_a^2 + \xi_b^2}/c^2) + \arctan(s_1 \sqrt{\xi_a^2 + \xi_b^2}/c)$$

*for some constant $\rho > 1$, then **wda-eig** converges linearly at the rate $\frac{1}{\rho}$.*

Corollary 1 implies that given a data matrix, the convergence rate of **WDA-eig** depends on the initialization, the subspace dimension $p$ and $\xi_a$, $\xi_b$. $\xi_a$ and $\xi_b$ are functions of $\xi^{c,c'}$ and depends on the Wasserstein regularization parameter $\lambda$. When $\lambda = 0$, $t^{c,c'}$ is a constant matrix and $\xi^{c,c'} = 0$. For a fixed $\lambda$, the arctangent gap $\eta$ depends on the inherent structure of the data matrix and whether a discriminative subspace exists. For example, given two clusters of data generated from 2D normal distributions as shown in Figure 1, $\eta$ depends on the separation of these two clusters. We can calculate $\eta^* \triangleq \mu_p^* - \mu_{p+1}^*$ since we know the true subspace $\mathbf{P}^*$, and we also run **WDA-eig** on a random initialization to get $\eta$. We observe that $\eta$ is close to 0 when the clusters overlap and is a monotonically increasing function of the Euclidean distance between the mean of the two clusters.

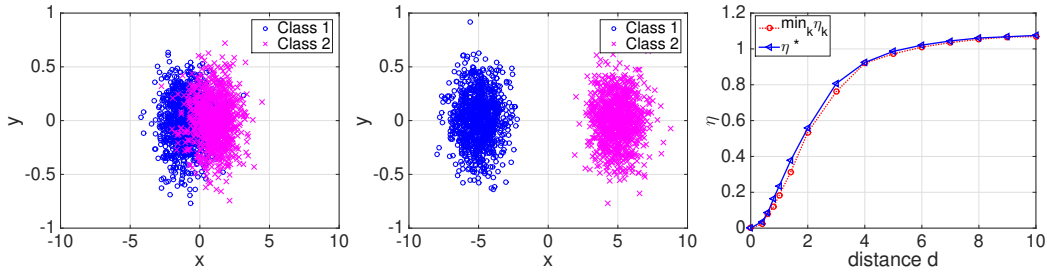

Figure 1: Left and middle: two classes of data generated from two random normal distributions: $X_i \, \mathcal{N} \in (\mu_i, \Sigma)$, $\mu = (\pm x, 0)$ where $x \in [0, 5]$. $x = 1$ in the Left and $x = 5$ in the Middle. Right: Arctangent gap $\eta$ as a function of the distance between the means $d \triangleq \| \mu_1 - \mu_2 \|$.

By applying the algorithm to a simulated dataset with 3 classes and 2 discriminative dimensions, we draw log plots of the distances between the subspaces subject to these components in Figure 2 to illustrate linear convergence rates. On the left we show $s_k$ with different values of the subspace dimension $p$ and with $\lambda = 0.1$ fixed. With $p = 2$ the algorithm achieves the fastest rate because the dimension of the true discriminative subspace is 2. In FDA, since $\mathbf{C}_b$ has rank $Nc - 1$ (where $Nc$ is the number of classes), $p$ has to be $\leq Nc - 1$. In **WDA-eig**, $p$ is less restrictive, but choosing $p \geq Nc$ may still slow down or prevent convergence if $\lambda$ is small. In the middle we show $s_k$ with different values of the Wasserstein regularizer $\lambda$ and with $p = 2$ fixed. When $\lambda$ is small, the matrices $\mathbf{C}_w$ and $\mathbf{C}_b$ in WDA can be viewed as the matrices in FDA with a small perturbation, and in such cases the Lipschitz constants $\xi_a$ and $\xi_b$ are close to zero so the algorithm is guaranteed to converge. We also observe that a larger $\lambda$ corresponds to a slower convergence rate. On the right we illustrate the effect of initialization for local convergence. We use the converged solution as an approximation to the true discriminative subspace $\mathbf{P}^*$ and plot the distance $\| \sin \Theta(\mathbf{P}^*, \mathbf{P}_{k-1}) \|$ for each iteration $k$, with varying $\hat{s}_0 = \| \sin \Theta(\mathbf{P}^*, \mathbf{P}_0) \|$. We observe that initialization has little effect on the convergence rate and that the algorithm converges in most cases except for the case where $\hat{s}_0 \approx 1$.

## 4  Numerical Experiments

In this section we evaluate the performance of the proposed Algorithm 1 on classification tasks by applying it to a simulated dataset and the MNIST dataset. We refer to our proposed algorithm as **WDA-eig** and refer to the original implementation in [12] with projected gradient descent as **WDA**. **WDA** converges when the norm of the gradient is below $10^{-6}$, and **WDA-eig** converges when the distance between two consecutive subspaces is less than $10^{-6}$.

### 4.1  Simulated dataset

We first compare **WDA-eig** with **WDA** on a simulated dataset. We use the same setup as given in [13], where the data belongs to 3 non-linearly separable classes and is generated using 2 discrim-

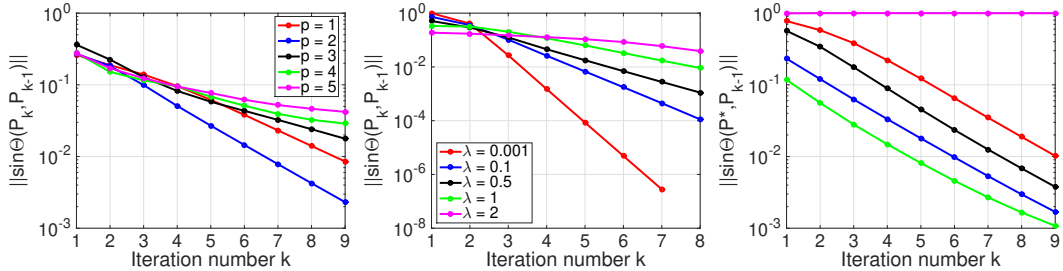

Figure 2: Left and Middle: Distances between subspaces $\|\sin\Theta(\mathbf{P}_k, \mathbf{P}_{k-1})\|$ as a function of iteration number $k$, with varying $\lambda$ and $p$, respectively. Right: $\|\sin\Theta(\mathbf{P}^*, \mathbf{P}_{k-1})\|$ as a function of iteration number $k$, with varying initialization $\mathbf{P}_0$.

Table 1: Comparison between **WDA** and **WDA-eig**

| Param $\lambda$ | Algo | Prob. of Convergence | Avg. Acc.(std.) | Converged Acc.(std.) | CPU time |
|---|---|---|---|---|---|
| $\lambda = 0.1$ | **wda** | 25% | 0.712(0.152) | 0.977(0) | 72 |
| | **wda-eig** | 100% | 0.968(0) | 0.968(0) | 0.677 |
| $\lambda = 1.0$ | **wda** | 78% | 0.908(0.149) | 0.987(0) | 6.37 |
| | **wda-eig** | 100% | 0.986(0) | 0.986(0) | 1.17 |
| $\lambda = 5.0$ | **wda** | 73% | 0.885(0.164) | 0.985(0) | 7.04 |
| | **wda-eig** | 100% | 0.985(0) | 0.985(0) | 1.61 |

inant features and 8 dimensions of Gaussian noise. We apply these two algorithms with varying regularization parameter $\lambda$, and compare their computational efficiency and classification accuracy with a K-Nearest-Neighbors classifier (KNN) on the projected data ($k = 10$). For each $\lambda$, we run each algorithms for 100 randomly-initialized trials, and the results are shown in Table 1. The third column of the table shows the probability of convergence over 100 trials, and the fourth column shows the accuracy averaged over trials. For $\lambda = 0.1, 1, 5$, **WDA-eig** converges in all the trials with zero standard deviations and achieves higher accuracy scores on average, while **WDA** has high standard deviation due to the low probability of convergence. The fifth column shows the accuracy averaged only for the converged trials, and **WDA-eig** and **WDA** have comparable performances in accuracy, which indicates that the ratio trace formulation can serve as a good approximation to the trace ratio formulation. The last column shows the efficiency measured by averaged CPU time in seconds over 100 trials. **WDA-eig** takes shorter running time than **WDA** since the former only requires a few iterations to converge and the running time per SCF iteration is comparable to the running time per gradient descent iteration. Even in cases where most trials converge for both solvers (e.g., when $\lambda = 1$), **WDA** takes more iterations to converge on average.

## 4.2 MNIST dataset

Next, we test the classification performance on a real dataset and also evaluate the generalization ability of the proposed approach. We extract 1000 samples in the MNIST dataset as the training set and use 10000 samples in the test set. We measure the KNN prediction error on the projected data as a function of the subspace dimension $p$, the number of nearest neighbors $K$, and the Wasserstein regularization parameter $\lambda$ respectively in Figure 3. On the left we show the prediction error of full data/PCA/FDA/**WDA**/**WDA-eig**+KNN applied to the original data as a function of $p$, with $\lambda = 0.01$ and $K = 10$ fixed. In implementation of FDA/**WDA-eig** we add a small perturbation term $\epsilon I_p$ on $\mathbf{C}_w$ to make the denominator positive definite, and we choose $\epsilon = 2$ in this setting, which removes the restriction of $p \leq 9$ for FDA. In the middle we show the performance of these methods as a function of $K$. Another approach to avoid $\mathbf{C}_w$ being semidefinite is to project away the null space of the data matrix before applying discriminant analysis. To achieve this end, we first apply PCA to the original data matrix and retain only the first 20 principal components. We then apply PCA/FDA/LFDA [37]/**WDA**/**WDA-eig** on the dimension-reduced data to obtain a subspace of dimension $p = 9$ without any regularization on $\mathbf{C}_w$, and the results are shown on the right.

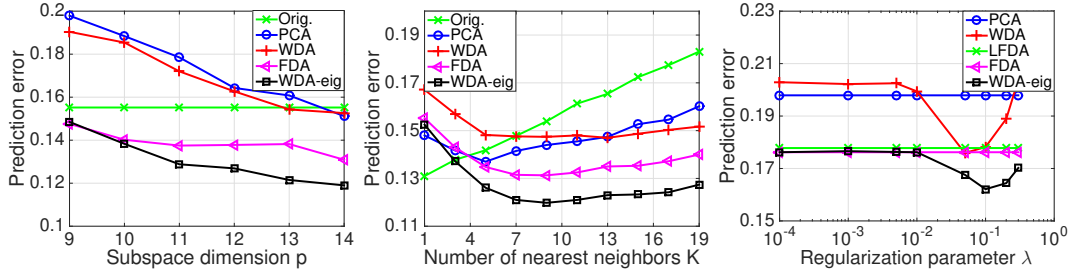

Figure 3: Prediction error as a function of the subspace dimension $p$, the number of nearest neighbors $K$, and the Wasserstein regularization parameter $\lambda$.

## 5 Unsupervised WDA

Since WDA is a dimensionality reduction technique, it could also be integrated with a low-dimensional clustering technique to do high-dimensional clustering. Here we propose Algorithm 2 to extend WDA to the unsupervised setting.

### 5.1 Clustering Algorithm

---
**Algorithm 2** Iterative WDA clustering
---
Input: De-meaned dataset $X$, $\mathbf{P}_0 \in \mathbb{O}^{d \times p}$, tolerance $\epsilon$, max number of iterations $N$
**for** $k = 0, 1, \cdots, N$ **do**
    Compute $y = X\mathbf{P}_k \in \mathbb{R}^{n \times p}$
    Cluster $y$ into $K$ classes and obtain the class labels $\hat{y}$
    Call Algorithm 1 to update $\mathbf{P}$ : $\mathbf{P}_{k+1} = \mathbf{WDA\text{-}eig}(X, \hat{y}, \mathbf{P}_k)$
    **if** the change in $\mathbf{P}_k$ is sufficiently small **then**
        Break
    **end if**
**end for**
---

We start with an initial guess and adaptively improve its labeling by performing clustering in the projected space. The goal is to converge to a discriminative subspace that will render the most accurate labels. Algorithm 2 solves the following optimization problem:

$$\max_{\mathbf{P}} \hat{J}(\mathbf{P}, \mathbf{T}, \hat{y}) \quad \text{s.t. } \mathbf{T}^{c,c'} = \underset{\mathbf{T} \in U_{n_c n_{c'}}}{\operatorname{argmin}} E_1(\mathbf{T}, \mathbf{P}, \hat{y}), \ \hat{y} = \underset{\hat{y}}{\operatorname{argmin}} E_2(\mathbf{P}, \hat{y}), \tag{8}$$

and $E_2$ is the objective of any specific low-dimensional clustering technique. The algorithm uses an alternating optimization scheme: for each iteration, given the class labels $\hat{y}$, $\mathbf{P}$ is chosen to maximize the ratio-trace problem $\hat{J}$, and then given the subspace, it finds the optimal labeling according to the clustering objective $E_2$. The objectives $E_2$ and $\hat{J}$ do not always align. A special case is FDA-Kmeans (or LDA-Kmeans) [10], where minimizing $E_2$ is equivalent to maximizing $\hat{J}$. It is derived that iteratively applying FDA and K-means is the same as alternating optimization in a unified objective [42], and that combining FDA and K-means is equivalent to kernel K-means in the original space with a specific kernel Gram matrix [48].

However, there is no theoretical guarantee that a larger objective value corresponds to a better clustering result in terms of external evaluation criteria. We observe that for FDA-Kmeans, the adjusted random index (ARI) [30] does not increase monotonically with the iteration number and could even converge to a worse result compared to the initial guess. For WDA, K-Means in the projected space does not maximize $\hat{J}$, but empirically we observe that several iterations with K-Means does improve clustering result in terms of external evaluation criteria such as ARI. Since the performance of FDA degrades when class distributions are multimodal, FDA could perform poorly given the wrong labels even if the true underlying distribution is Gaussian. On the other hand, we numerically observe that WDA is more robust to noisy labels due to a balance of local and global information (see supplementary material).

Table 2: Clustering results.

| Method | Dataset | ARI | NMI | Homogeneity | Completeness | FMI |
|---|---|---|---|---|---|---|
| Baseline | MNIST | $0.334 \pm 0.007$ | $0.475 \pm 0.006$ | $0.473 \pm 0.006$ | $0.477 \pm 0.007$ | $0.403 \pm 0.007$ |
| PCAKm | MNIST | $0.334 \pm 0.005$ | $0.471 \pm 0.005$ | $0.469 \pm 0.004$ | $0.473 \pm 0.007$ | $0.402 \pm 0.005$ |
| FDAKm | MNIST | $0.360 \pm 0.006$ | $0.500 \pm 0.007$ | $0.497 \pm 0.006$ | $0.503 \pm 0.008$ | $0.426 \pm 0.005$ |
| WDAKm | MNIST | $\mathbf{0.398 \pm 0.008}$ | $\mathbf{0.526 \pm 0.006}$ | $\mathbf{0.524 \pm 0.006}$ | $\mathbf{0.528 \pm 0.006}$ | $\mathbf{0.459 \pm 0.008}$ |
| Baseline | KTH | $0.424 \pm 0.035$ | $0.576 \pm 0.035$ | $0.556 \pm 0.035$ | $0.598 \pm 0.033$ | $0.535 \pm 0.028$ |
| PCAKm | KTH | $0.470 \pm 0.017$ | $0.616 \pm 0.009$ | $0.596 \pm 0.011$ | $0.637 \pm 0.007$ | $0.571 \pm 0.011$ |
| FDAKm | KTH | $0.481 \pm 0.022$ | $0.635 \pm 0.018$ | $0.614 \pm 0.019$ | $0.657 \pm 0.017$ | $0.580 \pm 0.016$ |
| WDAKm | KTH | $\mathbf{0.488 \pm 0.020}$ | $\mathbf{0.643 \pm 0.015}$ | $\mathbf{0.623 \pm 0.016}$ | $\mathbf{0.663 \pm 0.014}$ | $\mathbf{0.584 \pm 0.014}$ |
| Baseline | 15scene | $0.161 \pm 0.009$ | $0.352 \pm 0.009$ | $0.336 \pm 0.009$ | $0.369 \pm 0.009$ | $0.233 \pm 0.008$ |
| PCAKm | 15scene | $0.160 \pm 0.007$ | $0.351 \pm 0.006$ | $0.334 \pm 0.007$ | $0.368 \pm 0.006$ | $0.232 \pm 0.005$ |
| FDAKm | 15scene | $0.150 \pm 0.009$ | $0.350 \pm 0.011$ | $0.324 \pm 0.010$ | $0.376 \pm 0.014$ | $0.234 \pm 0.010$ |
| WDAKm | 15scene | $\mathbf{0.170 \pm 0.010}$ | $\mathbf{0.366 \pm 0.012}$ | $\mathbf{0.350 \pm 0.011}$ | $\mathbf{0.384 \pm 0.012}$ | $\mathbf{0.243 \pm 0.009}$ |
| Baseline | 20ng | $0.081 \pm 0.011$ | $0.235 \pm 0.021$ | $0.217 \pm 0.020$ | $0.255 \pm 0.023$ | $0.151 \pm 0.009$ |
| PCAKm | 20ng | $0.097 \pm 0.003$ | $0.247 \pm 0.005$ | $0.238 \pm 0.005$ | $0.255 \pm 0.005$ | $0.152 \pm 0.003$ |
| FDAKm | 20ng | $0.113 \pm 0.013$ | $0.298 \pm 0.019$ | $0.275 \pm 0.019$ | $0.322 \pm 0.020$ | $0.185 \pm 0.010$ |
| WDAKm | 20ng | $\mathbf{0.128 \pm 0.011}$ | $\mathbf{0.302 \pm 0.014}$ | $\mathbf{0.283 \pm 0.013}$ | $\mathbf{0.322 \pm 0.015}$ | $\mathbf{0.194 \pm 0.010}$ |

## 5.2 Experiments on WDA Clustering

In this section we evaluate the proposed Algorithm 2 and compare with other subspace clustering techniques. In what follows, let $Nc$ denote the number of classes, $n$ be the number of observations and $d$ be the number of features.

We use four real world datasets to evaluate the proposed method: the MNIST dataset for digits recognition, the 15-scene dataset [19] for multi-class image recognition, the KTH action recognition database [33] for multi-class video recognition, and the 20 newsgroup dataset for text classification. To avoid the singularity of the $\mathbf{C}_w$ matrix in FDA and WDA, we first do a dimension reduction on the original dataset using PCA and retain the first $2 \times Nc$ principal components. We refer to this data as the dimension-reduced data. We apply four different clustering methods to the four dataset: (1) K-means on the original data (Baseline); (2) K-means on dimension-reduced data (PCAKm); (3) FDA-Kmeans (FDAKm) on the dimension-reduced data; (4) WDA-kmeans (Algorithm 2 combined with K-means) (WDAKm) on the dimension-reduced data. For (3) and (4) we use the subspace obtained by PCA as initialization and $p = Nc-1$ as the subspace dimensions. No regularization term is added to $\mathbf{C}_w$. The Wasserstein regularizer $\lambda$ is coarsely tuned, where we choose $\lambda = 0.01$ for MNIST and 15-scene, $\lambda = 10$ for KTH, and $\lambda = 5$ for 20ng. The results are averaged over 20 trials. We use five external evaluation criteria to evaluate the quality of the clustering solutions [30, 36, 31, 14]. The results in Table 2 show that WDAKm achieves the best performance on all four datasets, in terms of the 5 external metrics we use. We also notice that in 15-scene dataset the performance of FDAKm is worse than the baseline method, which means with some wrong tags, FDA tends to overly separate data and decrease the clustering quality. In contrast, WDA always improve the clustering even in the difficult case such as the 15-scene dataset.

## 6 Conclusion

In this paper, we present a ratio trace formulation of the Wasserstein Discriminant Analysis and an eigensolver-based algorithm: **WDA-eig** to solve the problem. Unlike the original trace ratio formulation, the ratio trace formulation has a closed-form solution that is readily obtainable by the generalized eigenvalue decomposition once the regularized optimal transport problem is solved. We give a convergent analysis for **WDA-eig** under the SCF framework and numerically test the efficiency and convergence properties of the proposed algorithm. Although **WDA-eig** solves a slightly different problem, the ratio trace formulation can be served as an efficient alternative for the trace ratio formulation of WDA. **WDA-eig** also takes less time to converge on average and is less sensitive to initialization and parameters compared to **WDA**. As a supervised dimensionality reduction technique, WDA can also be combined with clustering techniques and applied iteratively to perform unsupervised learning. Numerical experiments show that the WDA clustering algorithm performs well on a set of real-world problems.

## Broader Impact

In the era of big data, business providers, data scientists, and governments try to explore opportunities in the large scale and high-dimensional datasets. Nevertheless, several major computational challenges arise and prevent practitioners from constructing effective algorithms or tools to analyze their datasets. Dimensionality reduction (DR) plays an essential role in supervised and unsupervised learning tasks when the datasets are high dimensional. One benefit of reducing the data dimension before classification or clustering is to save storage and reduce computational cost for the later steps, however, the DR technique itself can be costly. We study a recently proposed and promising DR technique, the Wasserstein discriminant analysis, and propose a different formulation that could achieve comparable or better results with less computational cost. We also analyze the problem from a different perspective that was originated from electronic structure calculations, which could be of interest to a broader audience in the machine learning community.

## Acknowledgments and Disclosure of Funding

We thank the anonymous Referees and Area Chair for their constructive comments. The work is supported by Baidu Research.

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
