[Supplementary Material]

# 7 Proof of Theorem 1. (Global Convergence of SCF)

Consider the generalized NLEP $A(\mathbf{P})\mathbf{V} = B(\mathbf{P})\mathbf{V}\Lambda$, where $\mathbf{V} = [v_1, \ldots, v_p]$ and $\mathbf{P}$ is an orthonormal basis of $\mathbf{V}$. $A(\mathbf{P})$, $B(\mathbf{P})$ are symmetric matrix-valued function and $B(\mathbf{P})$ is positive definite. $\Lambda = \text{diag}(\lambda_1, \ldots, \lambda_p)$, where $\lambda_1 \geq \cdots \geq \lambda_p$ are the $p$ largest eigenvalues of $(A(\mathbf{P}), B(\mathbf{P}))$ corresponding to eigenvectors $v_1, \ldots, v_p$. We emphasize that $A(\mathbf{P})$, $B(\mathbf{P})$ are invariant to orthogonal transformation of $\mathbf{P}$, i.e., $A(\mathbf{P}) \equiv A(\mathbf{P}Q)$, $B(\mathbf{P}) \equiv B(\mathbf{P}Q)$ for any orthogonal matrix $Q \in \mathbb{R}^{p \times p}$.

**Definitions.** Let $\mathcal{X}$ and $\mathcal{Y}$ be two $p$-dimensional subspaces of $\mathbb{R}^n$. Let the columns of $X$ form an orthonormal basis for $\mathcal{X}$ and the columns of $Y$ form an orthonormal basis for $\mathcal{Y}$. We use $\|\sin\Theta(\mathcal{X}, \mathcal{Y})\|$ as in [35] to measure the distance between $\mathcal{X}$ and $\mathcal{Y}$, where

$$\Theta(\mathcal{X}, \mathcal{Y}) = \text{diag}(\theta_1(\mathcal{X}, \mathcal{Y}), \ldots, \theta_p(\mathcal{X}, \mathcal{Y})).$$

Here, $\theta_j(\mathcal{X}, \mathcal{Y})$'s denote the *canonical angles* between $\mathcal{X}$ and $\mathcal{Y}$ [p. 43][35], which is defined as

$$0 \leq \theta_j(\mathcal{X}, \mathcal{Y}) \triangleq \arccos\sigma_j \leq \frac{\pi}{2} \quad \text{for } 1 \leq j \leq k,$$

where $\sigma_j$'s are the singular values of $X^T Y$. Similar to the Crawford number for symmetric definite matrix pair $(A, B)$ [Chapter 8.7] [40], we define the Crawford number for the generalized NLEP as

$$c \triangleq \min_{\mathbf{P} \in \mathbb{O}^{d \times p}} \min_{x \in \mathbb{C}^d, \|x\|=1} (x^T (A(\mathbf{P}) + iB(\mathbf{P}))x).$$

Define $C \triangleq \max_{\mathbf{P} \in \mathbb{O}^{d \times p}} \sqrt{\|A(\mathbf{P})^2 + B(\mathbf{P})^2\|}$. At the $k$th SCF iteration, one computes an approximation to the eigenvector matrix $\mathbf{V}_k$ associated with the $p$ largest eigenvalues of $(A(\mathbf{P}_{k-1}), B(\mathbf{P}_{k-1}))$, where $\mathbf{P}_{k-1}$ is an orthonormal basis for $\mathbf{V}_{k-1}$, and then $\mathbf{V}_k$ is used as the next approximation to the solution. Let $A_k = A(\mathbf{P}_k)$, $B_k = B(\mathbf{P}_k)$, $\mu_{i,k} = \arctan\lambda_i(A(\mathbf{P}_k), B(\mathbf{P}_k))$.

We study the convergence of SCF iteration under the following assumptions:

**A1:** For any $\mathbf{P}_1, \mathbf{P}_2 \in \mathbb{O}^{d \times p}$, assume that there exist positive constants $\xi_a, \xi_b$ such that

$$\|A(\mathbf{P}_1) - A(\mathbf{P}_2)\| \leq \xi_a \|\sin\Theta(\mathbf{P}_1, \mathbf{P}_2)\|, \quad \|B(\mathbf{P}_1) - B(\mathbf{P}_2)\| \leq \xi_b \|\sin\Theta(\mathbf{P}_1, \mathbf{P}_2)\|;$$

**A2:** For $k = 1, 2, \cdots$, there exists an $\eta > 0$ such that

$$\mu_{p,k} - \mu_{p+1,k} \geq \eta.$$

**Theorem 1.** *(Global Convergence)* Let $s_1 = \|\sin\Theta(\mathbf{P}_0, \mathbf{P}_1)\|$. *Assume A1 and A2, and* $s_1\sqrt{\xi_a^2 + \xi_b^2} < c$. *If*

$$\eta > \arcsin(\rho C\sqrt{\xi_a^2 + \xi_b^2}/c^2) + \arctan(s_1\sqrt{\xi_a^2 + \xi_b^2}/c)$$

*for some constant $\rho > 1$, then SCF converges linearly at the rate of $\frac{1}{\rho}$.*

In order to show Theorem 1, we need the following three lemmas. The first lemma gives some fundamental results for $\|\sin\Theta(X, Y)\|$, which can be verified via definition.

**Lemma 1.** *Let $[X, X_c]$ and $[Y, Y_c]$ be two orthogonal matrices with $X, Y \in \mathbb{R}^{n \times k}$. Then*

$$\|\sin\Theta(X, Y)\| = \|X_c^T Y\| = \|X^T Y_c\| = \|XX^T - YY^T\|.$$

The next lemma gives perturbation bound for the eigenvalues of definite matrix pair.

**Lemma 2.** *[Theorem 8.7.3][40] Let $A, B$ be symmetric, $B$ be positive definite. Let the eigenvalues of $(A, B)$ be $\lambda_1 \geq \cdots \geq \lambda_n$. Let $c(A, B)$ be the Crawford number of $\{A, B\}$:*

$$c(A, B) \equiv \min_{x \in \mathbb{C}^n, \|x\|=1} |x^T (A + iB)x|.$$

*Suppose E and F are symmetric matrices that satisfy*

$$\epsilon^2 = \|E\|^2 + \|F\|^2 < c^2(A, B).$$

*Then $B + F$ is positive definite, and the eigenvalues $\tilde{\lambda}_1 \geq \ldots \tilde{\lambda}_n$ of $(A + E, B + F)$ satisfy*

$$|\arctan \tilde{\lambda}_i - \arctan \lambda_i| \leq \arctan \frac{\epsilon}{c(A, B)}, \ \forall 1 \leq i \leq n.$$

The following lemma gives perturbation bound for the eigenspace of definite matrix pair, which is rewritten from [Theorem 2.1] [38].

**Lemma 3.** *Let $A$, $B$, $\widetilde{A}$, $\widetilde{B}$ be symmetric, $B$ and $\widetilde{B}$ be positive definite. Let the eigenvalues of $(A, B)$ and $(\widetilde{A}, \widetilde{B})$ be $\tan \mu_1 \geq \cdots \geq \tan \mu_n$, $\tan \tilde{\mu}_1 \geq \cdots \geq \tan \tilde{\mu}_n$, respectively, the corresponding eigenvectors be $v_1, \ldots, v_n$, and $\tilde{v}_1, \ldots, \tilde{v}_n$, respectively. Assume that there are $\alpha \geq 0$ and $\delta > 0$ satisfying $\alpha + \delta \leq 1$, and a real number $\gamma$ such that*

$$|\sin(\gamma - \mu_i)| \leq \alpha, \ for \ i = 1, \ldots, p,$$
$$|\sin(\gamma - \tilde{\mu}_j)| \geq \alpha + \delta, \ for \ j = p + 1, \ldots, n$$

*(or vice-versa). Let $V_1 = [v_1, \ldots, v_p]$, $\widetilde{V}_1 = [\tilde{v}_1, \ldots, \tilde{v}_p]$. Then*

$$\|\sin \Theta(V_1, \widetilde{V}_1)\| \leq \frac{p(\alpha, \delta; \gamma)\sqrt{\|A^2 + B^2\|}}{c(A, B)c(\widetilde{A}, \widetilde{B})} \times \frac{\sqrt{\|(\widetilde{A} - A)^2 + (\widetilde{B} - B)^2\|}}{\delta},$$

*where*

$$p(\alpha, \delta; \gamma) \triangleq \frac{q(\gamma)(\alpha + \delta)\sqrt{1 - \alpha^2} + \alpha\sqrt{1 - (\alpha + \delta)^2}}{2\alpha + \delta},$$

*with $q(\gamma) = \sqrt{2}$ for $\gamma \neq 0$ and $q(0) = 1$.*

Now we are ready to prove **Theorem 1**.

*Proof of Theorem 1.* Denote $A_k = A(\mathbf{P}_k)$, $B_k = B(\mathbf{P}_k)$, $E_k = A_k - A_{k-1}$, $F_k = B_k - B_{k-1}$. Without loss of generality, we assume that $A_k$ is also positive definite. Otherwise, we let $A_k = A_k + tB_k$ for sufficiently large $t$, then $A_k$ is positive definite and the sequence $\{\mathbf{V}_k\}$ produced by SCF iteration remains unchanged. Let $\Lambda_k = \mathrm{diag}(\lambda_{1,k}, \ldots, \lambda_{p,k})$, $\mathbf{V}_k = [v_{1,k}, \ldots, v_{p,k}]$, where $\lambda_{i,k}$ is the $i^{th}$ largest eigenvalue of $(A_k, B_k)$, $v_{i,k}$ is the corresponding eigenvector. Also denote $s_k = \|\sin \Theta(\mathbf{P}_k, \mathbf{P}_{k-1})\| = \|\mathbf{P}_k\mathbf{P}_k^T - \mathbf{P}_{k-1}\mathbf{P}_{k-1}^T\|$ as the distance between subspaces.

By assumption **A1**, we have

$$\sqrt{\|A_k - A_{k-1}\|^2 + \|B_k - B_{k-1}\|^2}$$
$$\leq \sqrt{\xi_a^2 + \xi_b^2}\|\sin \Theta(\mathbf{P}_k, \mathbf{P}_{k-1})\| = s_k\sqrt{\xi_a^2 + \xi_b^2}.$$

Now consider $k = 1$. By assumption, $s_1\sqrt{\xi_a^2 + \xi_b^2} < c$, then we may apply Lemma 2, which gives

$$|\mu_{i,1} - \mu_{i,0}| \leq \arctan \frac{\sqrt{\|A_1 - A_0\|^2 + \|B_1 - B_0\|^2}}{c}$$
$$\leq \arctan(s_1\sqrt{\xi_a^2 + \xi_b^2}/c), \ \forall 1 \leq i \leq n.$$

It follows that

$$\mu_{p,1} - \mu_{p+1,0} = \mu_{p,1} - \mu_{p+1,1} + \mu_{p+1,1} - \mu_{p+1,0}$$
$$\geq \eta - \arctan(s_1\sqrt{\xi_a^2 + \xi_b^2}/c)$$
$$> \arcsin(\rho C\sqrt{\xi_a^2 + \xi_b^2}/c^2) \geq 0, \tag{9}$$

where the last inequality uses the assumption

$$\eta > \arcsin(\rho C\sqrt{\xi_a^2 + \xi_b^2}/c^2) + \arctan(s_1\sqrt{\xi_a^2 + \xi_b^2}/c).$$

Now let

$$\gamma = \frac{\mu_{1,1} + \mu_{p,1}}{2}, \ \alpha = \sin\frac{\mu_{1,1} - \mu_{p,1}}{2}, \ \alpha + \delta = \sin(\frac{\mu_{1,1} + \mu_{p,1}}{2} - \mu_{p+1,0}),$$

then for all $1 \le i \le p$ and $p + 1 \le j \le n$, we have

$$|\sin(\gamma - \mu_{i,1})| \le |\sin\frac{\mu_{1,1} - \mu_{p,1}}{2}| = \alpha, \tag{10a}$$

$$|\sin(\gamma - \mu_{j,0})| \ge |\sin(\frac{\mu_{1,1} + \mu_{p,1}}{2} - \mu_{p+1,0})| = \alpha + \delta, \tag{10b}$$

$$\alpha + \delta \le 1, \qquad \gamma > 0, \tag{10c}$$

$$\delta = \sin(\frac{\mu_{1,1} + \mu_{p,1}}{2} - \mu_{p+1,0}) - \sin\frac{\mu_{1,1} - \mu_{p,1}}{2}$$

$$= 2\cos\frac{\mu_{1,1} - \mu_{p+1,0}}{2}\sin\frac{\mu_{p,1} - \mu_{p+1,0}}{2}. \tag{10d}$$

By calculations, we obtain

$$\begin{aligned}
p(\alpha, \delta; \gamma) &= \frac{(\alpha + \delta)\sqrt{1 - \alpha^2} + \alpha\sqrt{1 - (\alpha + \delta)^2}}{2\alpha + \delta}\\
&= \frac{\sin\frac{\mu_{1,1} - \mu_{p,1}}{2}\cos(\frac{\mu_{1,1} + \mu_{p,1}}{2} - \mu_{p+1,0}) + \cos\frac{\mu_{1,1} - \mu_{p,1}}{2}\sin(\frac{\mu_{1,1} + \mu_{p,1}}{2} - \mu_{p+1,0})}{\sin\frac{\mu_{1,1} - \mu_{p,1}}{2} + \sin(\frac{\mu_{1,1} + \mu_{p,1}}{2} - \mu_{p+1,0})}\\
&= \frac{\sin(\mu_{1,1} - \mu_{p+1,0})}{\sin\frac{\mu_{1,1} - \mu_{p,1}}{2} + \sin(\frac{\mu_{1,1} + \mu_{p,1}}{2} - \mu_{p+1,0})}\\
&= \frac{2\sin\frac{\mu_{1,1} - \mu_{p+1,0}}{2}\cos\frac{\mu_{1,1} - \mu_{p+1,0}}{2}}{2\sin\frac{\mu_{1,1} - \mu_{p+1,0}}{2}\cos\frac{\mu_{p,1} - \mu_{p+1,0}}{2}}\\
&= \frac{\cos\frac{\mu_{1,1} - \mu_{p+1,0}}{2}}{\cos\frac{\mu_{p,1} - \mu_{p+1,0}}{2}}. \tag{11}
\end{aligned}$$

Using Lemma 3, we have

$$s_2 \le \frac{p(\alpha, \delta; \gamma)C}{c^2} \cdot \frac{\sqrt{\|(A_1 - A_0)^2 + (B_1 - B_0)^2\|}}{\delta} \le \frac{p(\alpha, \delta; \gamma)C}{c^2} \cdot \frac{\sqrt{\xi_a^2 + \xi_b^2}}{\delta}s_1. \tag{12}$$

Substituting (10d), (11) into (12), and using (9), we have

$$s_2 \le \frac{1}{\rho}s_1. \tag{13}$$

where

$$\rho = \frac{c^2\sin(\mu_{p,1} - \mu_{p+1,0})}{C\sqrt{\xi_a^2 + \xi_b^2}} > 1. \tag{14}$$

For general $k = 2$, noticing the following holds

$$\eta > \arcsin(\rho C\sqrt{\xi_a^2 + \xi_b^2}/c^2) + \arctan(s_1\sqrt{\xi_a^2 + \xi_b^2}/c)$$

$$> \arcsin(\rho C\sqrt{\xi_a^2 + \xi_b^2}/c^2) + \arctan(s_2\sqrt{\xi_a^2 + \xi_b^2}/c).$$

Similar to the proof for $k = 1$, we can conclude $s_3 \le \frac{1}{\rho}s_2$. By induction, $s_{k+1} \le \frac{1}{\rho}s_k$, thus completing the proof. $\qquad\square$

# 8 Proof of Theorem 2. (Local Convergence of SCF)

With relaxed assumptions, we can show local convergence if the initial subspace is close enough to the true subspace $\mathbf{P}_*$:

**A3:** Let $\mu_i^*$ denote $\arctan \lambda_i(A(\mathbf{P}^*), B(\mathbf{P}^*))$. There exists an $\eta > 0$ such that

$$\mu_p^* - \mu_{p+1}^* \geq \eta.$$

**Theorem 2.** *(Local Convergence) Let* $\hat{s}_0 = \|\sin\Theta(\boldsymbol{P}_0, \boldsymbol{P}^*)\|$. *Assume A1 and A3, and* $\hat{s}_0\sqrt{\xi_a^2 + \xi_b^2} < c$. *If*

$$\eta > \arcsin(\rho C\sqrt{\xi_a^2 + \xi_b^2}/c^2) + \arctan(\hat{s}_0\sqrt{\xi_a^2 + \xi_b^2}/c)$$

*for some constant* $\rho > 1$, *then SCF is locally convergent at* $\boldsymbol{P}^*$ *at the rate of* $\frac{1}{\rho}$.

*Proof.* By assumption **A1**, we have

$$\sqrt{\|A_0 - A^*\|^2 + \|B_0 - B^*\|^2} \leq \sqrt{\xi_a^2 + \xi_b^2}\|\sin\Theta(\mathbf{P}_0, \mathbf{P}^*)\| = \hat{s}_0\sqrt{\xi_a^2 + \xi_b^2}.$$

Applying Lemma 2, we have

$$|\mu_{p,0} - \mu_p^*| \leq \arctan\frac{\sqrt{\|A_0 - A^*\|^2 + \|B_0 - B^*\|^2}}{c}$$

$$\leq \arctan(\hat{s}_0\sqrt{\xi_a^2 + \xi_b^2}/c), \ \forall 1 \leq p \leq n.$$

By assumption **A3** it follows that

$$\mu_{p,0} - \mu_{p+1}^* = \mu_{p,0} - \mu_p^* + \mu_p^* - \mu_{p+1}^* \geq \eta - \arctan(\hat{s}_0\sqrt{\xi_a^2 + \xi_b^2}/c)$$

$$> \arcsin(\rho C\sqrt{\xi_a^2 + \xi_b^2}/c^2) \geq 0. \tag{15}$$

Following the same procedures as in the proof for **Theorem 1**, we arrive that

$$\|\sin\Theta(\mathbf{P}_{k-1}, \mathbf{P}^*)\| \leq \frac{1}{\rho}\|\sin\Theta(\mathbf{P}_k, \mathbf{P}^*)\|,$$

where $\rho = \frac{c^2 \sin(\mu_{p,0} - \mu_{p+1}^*)}{C\sqrt{\xi_a^2 + \xi_b^2}}$. $\qquad\square$

# 9 Proof of Corollary 1. (Convergence of WDA-eig)

**Corollary 1.** *Suppose that the optimal transport matrix $\boldsymbol{T}^{c,c'}$ satisfies a Lipschitz-like condition:*

$$\|\boldsymbol{T}^{c,c'}(\boldsymbol{P}_1) - \boldsymbol{T}^{c,c'}(\boldsymbol{P}_2)\| \leq \xi^{c,c'} \|\sin\Theta(\boldsymbol{P}_1, \boldsymbol{P}_2)\|,$$

*For a given $p$, let*

$$\eta = \min_k \{\mu_{p,k} - \mu_{p+1,k}\}.$$

*Denote $\xi_a = \sum_{c,c'>c} \xi^{c,c'} \|\sum_{i,j}(x_i^c - x_j^{c'})(x_i^c - x_j^{c'})^T\|$, $\xi_b = \sum_c \xi^{c,c} \|\sum_{i,j}(x_i^c - x_j^c)(x_i^c - x_j^c)^T\|$. If*

$$\eta > \arcsin(\rho C\sqrt{\xi_a^2 + \xi_b^2}/c^2) + \arctan(s_1\sqrt{\xi_a^2 + \xi_b^2}/c)$$

*for some constant $\rho > 1$, then **wda-eig** converges linearly at the rate $\frac{1}{\rho}$.*

*Proof.* We first note that in **WDA-eig**, $\mathbf{C}_b$ and $\mathbf{C}_w$ are invariant to orthogonal transformation of $\mathbf{P}$, i.e., $\mathbf{C}_b(\mathbf{P}) \equiv \mathbf{C}_b(\mathbf{P}Q)$, $\mathbf{C}_w(\mathbf{P}) \equiv \mathbf{C}_w(\mathbf{P}Q)$ for any orthogonal matrix $Q \in \mathbb{R}^{p \times p}$, since

$$\mathbf{C}_b(\mathbf{P}) = \sum_{c,c'>c} \sum_{i,j} t_{ij}^{c,c'}(\mathbf{P})(x_i^c - x_j^{c'})(x_i^c - x_j^{c'})^T,$$

$$\mathbf{C}_w(\mathbf{P}) = \sum_c \sum_{i,j} t_{ij}^{c,c}(\mathbf{P})(x_i^c - x_j^c)(x_i^c - x_j^c)^T,$$

and $\mathbf{T}^{c,c'}(\mathbf{P})$ and $\mathbf{T}^c(\mathbf{P})$ are invariant to orthogonal transformation of $\mathbf{P}$:

$$\mathbf{T}^{c,c'}(\mathbf{P}Q) \triangleq \underset{\mathbf{T} \in U_{n_c n_{c'}}}{\operatorname{argmin}} \lambda\langle \mathbf{T}, \mathbf{M}_{X^c \mathbf{P}Q, X^{c'}\mathbf{P}Q}\rangle + \sum_{i,j} t_{ij}\log(t_{ij})$$

$$= \underset{\mathbf{T} \in U_{n_c n_{c'}}}{\operatorname{argmin}} \lambda\sum_{i,j} t_{ij}\|(x_i^c - x_j^{c'})^T \mathbf{P}Q\| + \sum_{i,j} t_{ij}\log(t_{ij})$$

$$= \underset{\mathbf{T} \in U_{n_c n_{c'}}}{\operatorname{argmin}} \lambda\sum_{i,j} t_{ij}\|(x_i^c - x_j^{c'})^T \mathbf{P}\| + \sum_{i,j} t_{ij}\log(t_{ij}) = \mathbf{T}^{c,c'}(\mathbf{P}).$$

By the assumption on $\mathbf{T}^{c,c'}$, $\mathbf{C}_b$ satisfies

$$\|\mathbf{C}_b(\mathbf{P}_1) - \mathbf{C}_b(\mathbf{P}_2)\| = \|\sum_{c,c'>c}\sum_{i,j}(t_{ij}^{c,c'}(\mathbf{P}_1) - t_{ij}^{c,c'}(\mathbf{P}_2))(x_i^c - x_j^{c'})(x_i^c - x_j^{c'})^T)\|$$

$$\leq \sum_{c,c'>c} \max_{i,j}|t_{ij}^{c,c'}(\mathbf{P}_1) - t_{ij}^{c,c'}(\mathbf{P}_2)|\|\sum_{i,j}(x_i^c - x_j^{c'})(x_i^c - x_j^{c'})^T\|$$

$$\leq \sum_{c,c'>c} \xi^{c,c'}\|\sin\Theta(\mathbf{P}_1,\mathbf{P}_2)\|\|\sum_{i,j}(x_i^c - x_j^{c'})(x_i^c - x_j^{c'})^T\|$$

$$\triangleq \xi_a\|\sin\Theta(\mathbf{P}_1,\mathbf{P}_2)\|.$$

The last inequality holds since

$$\max_{i,j}|t_{ij}^{c,c'}(\mathbf{P}_1) - t_{ij}^{c,c'}(\mathbf{P}_2)| \leq \|\mathbf{T}^{c,c'}(\mathbf{P}_1) - \mathbf{T}^{c,c'}(\mathbf{P}_2)\| \leq \xi^{c,c'}\|\sin\Theta(\mathbf{P}_1,\mathbf{P}_2)\|.$$

Similarly,

$$\|\mathbf{C}_w(\mathbf{P}_1) - \mathbf{C}_w(\mathbf{P}_2)\| \leq \sum_c \xi^{c,c}\|\sum_{i,j}(x_i^c - x_j^c)(x_i^c - x_j^c)^T\|\|\sin\Theta(\mathbf{P}_1,\mathbf{P}_2)\| \triangleq \xi_b\|\sin\Theta(\mathbf{P}_1,\mathbf{P}_2)\|.$$

For all iteration number $k$, $\mu_{p,k} - \mu_{p+1,k} \geq \eta$. Since **A1** and **A2** are satisfied, the result follows directly from **Theorem 1**. $\square$

# 10  Sensitivity to Noisy Labels

For iterative subspace clustering, performing K-Means on the projected data may not render accurate labels in the first few iterations, especially if we initialize with random subspace. We therefore investigate how the subspace changes when we perturb the labels. The results in Table 3 illustrate the sensitivity to noisy labels of FDA (same as **WDA-eig** with $\lambda = 0$), **WDA-eig** ($\lambda = 1.0$) and local FDA (LFDA) [37] with the number of neighbors$= 1$. We use the simulated dataset introduced in the Main Paper, Section 4.1 and add noisy labels to the data. The first column of Table 3 shows the percentage of wrong labels added. The rest of the columns show the distance of the subspace $\mathbf{P}$ obtained by FDA/**WDA-eig**/LFDA under the noisy labels to the original subspaces $\mathbf{P}^*$ measured by $\| \sin \Theta(\mathbf{P}, \mathbf{P}^*) \|$, where the original subspaces are approximated by the converged solution of FDA/**WDA-eig**/LFDA under true labels. The results are averaged over 20 trials. We observe that WDA is more robust to noisy labels than both FDA and local FDA.

Table 3: Sensitivity to Noisy Labels.

| % wrong labels | FDA dist. to $\mathbf{P}^*$ | WDA dist. to $\mathbf{P}^*$ | LFDA dist. to $\mathbf{P}^*$ |
|:---:|:---:|:---:|:---:|
| 1% | 0.21 | **0.01** | 0.04 |
| 5% | 0.32 | **0.02** | 0.08 |
| 10% | 0.59 | **0.05** | 0.11 |
| 20% | 0.84 | **0.07** | 0.15 |