[Reviews · NeurIPS 2020]

Review 1

Summary and Contributions: Update after discussion period: I thank the authors for their response, which addresses some of my comments. I still think that the motivation for switching objectives could be made more crisp and intuitive, and that there’s still some questions about how realistic the assumptions are. So I would not increase my score, but I certainly have no reason to decrease it either, as I would like to see the paper accepted. ----------- This paper proposes a new approach to solve the Wasserstein Discriminant Analysis (WDA) problem, which is a recent Optimal Transport-based dimensionality reduction technique. The crux of the approach proposed here is replacing the “trace-ratio” formulation of WDA for a “ratio-trace” one, an approximation that is common in numerical analysis. Albeit simple, this reformulation leads down a distinct optimization route (a non-linear eigensolver via self-consistent iterations), allowing the authors to provide clean convergence results, and yielding improved convergence in practice. The approach is instantiated in two empirical settings, clustering and classification, and it is show to perform favorably to other similar baselines.

Strengths: - Relevant problem. Although narrowly focused, the scope of this paper is a problem of significant relevance to the ML community - Sound/elegant/novel approach. Despite the apparent subtlety of the change in formulation (trace-ratio -> ratio-trace), the paper manages to come up with a sound and elegant approach that makes this new formulation compelling

Weaknesses: - Empirical evaluation is somewhat limited, misses some baselines, and it one instance makes claims on results that are well within margins of error - Some minor issues with writing and clarity, in particular missing discussion of assumptions (see below)

Correctness: The overall approach is sound and the theoretical results seem correct. I only have a few comments about framing/presentation in this regard: * The paper hinges on a trace-ratio -> ratio-trace substitution, but provides little intuition about this, instead seemingly assuming the reader is familiar with such a trick (I was not). It would be useful to motivate this better (e.g., why is this typically done? afterall, replacing scalar reciprocals by matrix inverses seems counterintuitive at first sight). Also, given that this is an approximation, I would have expected some discussion on the *quality* of the approximation (can it be quantified?), and whether it still captures the relevant aspects of the problem it is approximating. Even further, the paper could provide some simulations comparing the solutions obtained with the two objectives (the analysis for Supervised WDA in Section 3.2 does not include classic WDA). * The ratio trace formulation involves an inverse of a term that might be singular (e.g. if rank Cw < p). What happens in that case? This doesn't seem to be discussed. * Some details and discussion are missing from the Analysis section. -> Early on, B(P) is assumed to be positive definite. What if it's semi-definite? -> What matrix norm is used in L132? -> There is barely (if any) discussion on the implications of assumptions A1,A2,A3. None of these are clear-cut, so I would have expected to see a discussion on the intuition behind these (e.g., A1 is a Lipschitz-type stability constraint on the A, B operators...), on how likely they are to be satisfied in practice. -> s_k is never defined as far as I can tell (it's ||sinΘ(Pk, Pk-1)|| it seems?) * In the experimental section 3.2, there is no comparison to the usual (trace ratio) WDA, which would have been very useful. * In L174-177, it says that small λ implies less perturbation and faster convergence rate. That seems counterintuitive with the definition in Eq (1), where small λ in fact implies *la

Clarity: Again, I have no major concerns about clarity either - the paper is easy to read, mostly clear. Some minor issues: * There's various typos throughout the paper. See section on additional feedback. * In Eq (1), the regularization parameter λ is on the OT term, which is the opposite of the usual formulation. Of course it's equivalent, but having lambda defined this way makes the interpretation confusing: large λ corresponds to *less* regularization, which not might be what the reader expects. Also, given that λ is used here for eigenvalues, I would suggest using another symbol for the regularization parameter. * In L168-170: "With p=2 the algorithm achieves the fastest rate ..." -> this statement is not clear, requires some more explanation * The caption of Fig 2 should probably spell out what those parameters are.

Relation to Prior Work: The paper does a pretty good job of discussing related work. If anything, a detailed in discussion on the conceptual difference between the objectives of WDA and WDA-eig would be useful, as discussed above.

Reproducibility: Yes

Additional Feedback: Typos/minor issues: - L25: guarantee -> guarantees - L94: The citation should be [11] not [10]. - L226: improves -> improve it? - L227: to an -> to a


Review 2

Summary and Contributions: The authors propose a ratio trace formulation of the initial trace ratio definition of WDA. This new formulation, albeit not equivalent to the initial one, admits a closed form solution for the subspace problem, making the whole computation easier to handle and more robust.

Strengths: This new formulation follows the classic way to transform difficult ratio trace problems into easier to solve trace ratio problems. This algorithm is the first stable algorithm to approximate WDA and yields both good experimental results and nice theoretical properties. The claims seem solid, the contribution is novel and improves dramatically upon the initial WDA formulation.

Weaknesses: I am no expert in this field and cannot comment upon the weaknesses of this work.

Correctness: I haven't checked the proofs and cannot guarantee their correctness.

Clarity: This paper is very well written.

Relation to Prior Work: Yes.

Reproducibility: Yes

Additional Feedback:


Review 3

Summary and Contributions: This paper proposes a new slightly modified formulation of WDA, which is easier to solve. I found the paper interesting and the contribution certainly useful. It is a nice improvement, since it removes inner non-convex iteration on the solver, but the overall problem is still non-convex and the convergence guarantees weak. I think it makes the method definitely simpler and more straightforward, but I am not sure it worth a full publication. I have read the rebuttal, which helped to clarify some of the raised issues. I decided to keep the same score.

Strengths: - This is a nice idea which makes WDA simpler and more tractable. - There is some theory of the convergence - There are extensive experiments, and an extension to unsupervised WDA

Weaknesses: - This is not anymore the same functional, and little is explained about why doing the change. - It is not clear whether practical improvement are due to the change of the function, or to the fact that the solvers is more efficient. - It is not clear to me whether the theory is relevant because the hypotheses seems hard to check, are not discussed, and the empirical experiment does not check whether the theory can be applied.

Correctness: Yes from a theoretical point of view, the paper is correct.

Clarity: I found that the introduction could be improved (see bellow).

Relation to Prior Work: The relation to previous work is made very clear.

Reproducibility: Yes

Additional Feedback: Here is a list of questions: - Beside proving an energy which is simpler to minimize, it is not clear the relation between the two loss functions. Section (2) does not provide any clue about why this makes sense, and does not really motivate the proposed method. - Is there some simple example where one can show that the proposed energy makes sense? Is better? - In the context of classical FDA, is it something already used? - Assumptions A1 and A2 are not discussed. I find it hard to believe that it is possible to ensure global linear convergence for such a non-convex problem. Are there cases where one can show that these assumptions hold? - The local convergence statement seems more realistic, and the authors show some numerical simulations suggesting that it holds in practice. But is this practical behavior explained by the theory? I.e. does A1 and A3 holds? Some remarks on the writing style: - The introduction uses mathematical notations without introducing them (such as A, B, P, T) which makes its very hard to understand without reading the whole paper. The author should probably consider re-writing the introduction, maybe first explaining the main idea without maths, then introducing precisely the concepts, and then stating the contributions and its relation to previous works. - The author should use two different notations instead of using J(P) in (3) and (4).


Review 4

Summary and Contributions: This paper derives a new formulation of the Wasserstein Discriminant Analysis (WDA) and proposes an eigensolver-based algorithm to compute the discriminative subspace. Also, convergence analysis is provided for the proposed algorithm under the self-consistent field framework. Experiments on real datasets are conducted to verify the effectiveness in both classification and clustering tasks.

Strengths: (1) a ratio trace formulation of the WDA problem is presented. (2) introducing the Fiteration to solve the problem, and provides a convergence analysis for the SCF framework.

Weaknesses: (1)More comparison algorithms are suggested to be involved for comprehensive evaluation, e.g. clustering algorithms based on KL-divergence metric. (2)As Wasserstein distance preserves the underlying geometry of the space, it is suggested that graph-structured data would be better for evaluating those algorithms.

Correctness: Yes

Clarity: Yes

Relation to Prior Work: Yes

Reproducibility: Yes

Additional Feedback: Authors must address the concerns in the part of "Weaknesses"

[Author Response · NeurIPS 2020]

We thank all reviewers for careful review and comments. We first address the questions that reviewers have in common:

1. Why ratio trace instead of trace ratio: Since WDA can be viewed as an extension to the classical Fisher linear discriminant analysis (FDA), we refer to the trace ratio and ratio trace formulations in the context of FDA. Statistically, these two formulations are both defined and are both served as criterion to maximize inter-class distance while minimizing intra-class distance (see [Fukunaga, 2013] page 446-447, eqn. (10.5) and (10.8)). When the reduced dimension $p = 1$, both the numerator and the denominator are scalars and these two formulations are equivalent. When $p > 1$, the ratio trace formulation iteratively finds $p$ orthonormal vectors $v_i$ to maximize $\frac{v_i^T A v_i}{v_i^T B v_i}$, while the trace ratio maximizes $\frac{\sum_{i=1}^p v_i^T A v_i}{\sum_{i=1}^p v_i^T B v_i}$. Both formulations are widely in the literature. For example, in our reference, [8, 41] used the ratio trace formulation of FDA while [14, 23] used the trace ratio formulation. Algebraically, these two formulations are not equivalent, and one is not upper/lower bounded by the other, so it is hard to quantify the difference. In an optimization sense, the ratio trace can be viewed as a relaxation to the trace ratio objective: the trace ratio problem is equivalent to the trace difference problem, which can be relaxed to the ratio trace problem.

2. What are the intuition behind assumptions A1, A2, A3, and why do they hold in practice: Since Theorem 1 and 2 essentially characterize how does the eigenspace vary when the matrix pair undergoes a small perturbation, the intuition behind assumptions A1 and A2 comes from matrix perturbation theory. When the matrices $A$ and $B$ are less sensitive to perturbation, the algorithm is easier to converge. This "sensitivity" is quantified in the Lipschitz constants $\xi_a, \xi_b$ in A1. A2 and A3 are relaxations to a stronger assumption that assumes that there is an arctan gap between the $p^{th}$ and $p + 1^{th}$ eigenvalue for $(A(P), B(P))$ constructed from any $P$, which guarantees that a discriminative subspace exists and is unique. For the toy example used to generate Figure 1 and Table 1 (described in Section 4.1), we know that the true discriminative subspace exists and has dimension 2, and we numerically checked that assumptions A1-A3 hold. For real datasets, whether the assumptions hold depends on the inherent structure of the data and specific choice of parameters, and the theory can provide some guidance in choosing the parameters $\lambda$, $p$ and initialization $P_0$. For example, we can start with a small $\lambda$ since it's easier to converge and adaptively increase $\lambda$. We can also initialize with the subspace found by FDA (append orthogonal columns if needed) because it is closer to the true subspace if $\lambda$ is small.

**Reviewer 1:** 1. What if the $C_w$ term is singular: There are two ways for circumventing this problem: first is to add a diagonal regularization term $\epsilon I$ on $C_w$ as we did in Line 213-219. By doing so we improve the numerical stability in computing the inverse. Another approach is to project away the null space as we discussed in Line 216-218.
2. Matrix 2-norm was used in Line 132. 3. Assumptions A1-A3: see the general comment 2 above.
4. Yes, $s_k$ is defined to be $\|\sin \Theta(P_k, P_{k-1})\|$. The definition was moved to the supplementary material.
5. Section 3.2 analyzed the convergence of WDA-eig under SCF framework. The usual WDA uses a gradient-based approach and has a different convergence criteria, so direct comparison between convergence curves may not be intuitive.
6. A small $\lambda$ indeed implies a larger regularization to the Wasserstein distance. Here, however, we are perturbing the covariance matrices in FDA and not perturbing the true Wasserstein distance. When $\lambda = 0$ only the regularization term remains and WDA is FDA. When $\lambda$ is small, WDA focuses more on global information and is more similar to FDA.

**Reviewer 3:** 1-3. Ratio trace iteratively finds directions that maximize the ratio of the inter-class and intra-class distance, while trace ratio maximizes total sum of inter-class distance while minimizing total sum of intra-class distance. In the context of classical FDA, these are two definitions that are commonly used in the literature. To our knowledge, there is no theoretical arguments showing one is strictly better than the other. See the general comment 1.
4. Global convergence: If the assumption A2 is made even stronger, we can show that the solution to the WDA problem exists and is unique, and the algorithm always converges to a global optimal (see the general comment 2). However, here we are not trying to prove that the algorithm converges to a global optimal, but rather, the algorithm converges to some point globally in a numerical sense, see Line 144-145.
5. Local convergence: If $\lambda$ is small, A1 always holds. From the perspective of numerical computation, A3 always holds. Furthermore, if the data are well-separated in the true discriminative subspace, $\eta$ in A3 should be large which yields faster convergence. We empirically observed linear convergence on toy example as well as on real datasets.
6. Writing styles: Yes, we agree that the mathematical concepts could be delayed to later sections.

**Reviewer 4:** Thanks for the suggestions. For the clustering algorithms, we emphasize that our work focuses on **finding a subspace** for high-dimensional data and can be combined with many other metrics to perform clustering. We have in fact performed other experiments where WDA is combined with other clustering methods such as spectral clustering and WDA is able to find a better subspace for clustering (KL divergence might need some tweak here since it is not symmetric and therefore is not strictly a metric). In the paper we intentionally selected WDA combined with K-Means as an example, because subspace clustering with K-means is studied extensively in the literature, but we can certainly include more in the future version. For data we used in our experiments, we just followed the standard experiments and datasets in the subspace clustering literature. Using graph-structured data that is more suitable for Wasserstein distance is an excellent insight and definitely worth pursuing in our future work.

[Meta-Review · NeurIPS 2020]

The reviewers appreciated the feedback and found the efficient solver to be of value to the NeurIPS community and recommended an accept. Note that there are several important comments (especially from R1 and R3) that have to be taken into account in the final version.